# Gut Microbiota and Autism: Unlocking Connections

**DOI:** 10.3390/nu17233706

**Published:** 2025-11-26

**Authors:** Valentina Biagioli, Mariarosaria Matera, Ilaria Cavecchia, Francesco Di Pierro, Nicola Zerbinati, Pasquale Striano

**Affiliations:** 1Microbiota International Clinical Society (MICS), 10123 Turin, Italy; 2Department of Neurosciences, Rehabilitation, Ophthalmology, Genetics, Maternal and Child Health, University of Genoa, 16126 Genova, Italy; 3Department of Pediatric Emergencies, Misericordia Hospital, 58100 Grosseto, Italy; 4Microbiomic Department, Koelliker Hospital, 10134 Turin, Italy; 5Scientific & Research Department, Velleja Research, 20125 Milan, Italy; 6Department of Medicine and Technological Innovation, University of Insubria, 21100 Varese, Italy; 7Pediatric Neurology and Muscular Diseases Unit, IRCCS Istituto “Giannina Gaslini”, 16147 Genova, Italy

**Keywords:** autism spectrum disorder, gut microbiota, microbiota–gut–brain axis, maternal microbiota, neurodevelopment, nutrition

## Abstract

**Background:** Autism Spectrum Disorder (ASD) is a multifactorial neurodevelopmental condition in which genetic predisposition interacts with environmental factors. Among these, the gut microbiota has emerged as a crucial modulator of the microbiota–gut–brain axis (MGBA), influencing neuroinflammation, neurotransmission, and behavior. This review aims to provide an updated and integrative overview of the relationship between gut microbiota, diet, and neurodevelopment in ASD. **Methods:** A comprehensive search was conducted in PubMed, Scopus, and Web of Science for articles published between 2010 and 2025. Original studies, systematic reviews, and meta-analyses in English were included. **Results:** Evidence from human and animal studies supports a strong association between gut dysbiosis and ASD-related behaviors. Alterations in microbial composition, characterized by reduced *Bifidobacterium* and *Prevotella* and increased *Clostridium* spp., have been linked to impaired intestinal barrier function, chronic inflammation, and altered production of microbial metabolites such as short-chain fatty acids and tryptophan derivatives. **Discussion:** Maternal dysbiosis, nutritional imbalances, and perinatal stressors may further modulate fetal neurodevelopment through immune and epigenetic pathways. Emerging data suggest that dietary modulation, targeted nutritional interventions, functional foods, prebiotics, probiotics, and postbiotics could help restore microbial balance and improve neurobehavioral outcomes. **Conclusions:** The gut microbiota represents a key biological interface between environment, metabolism, and neurodevelopment. It is, therefore, necessary to transform current knowledge about the microbiota and neurodevelopment into clinical, social, and health actions that offer real solutions to people with ASD and their families. From this perspective, focusing on prevention, promoting healthy lifestyles, and integrating new technologies represent the true tools for building a more sustainable and inclusive healthcare system.

## 1. Introduction

Autism Spectrum Disorders (ASDs) represent some of the most complex and heterogeneous neurodevelopmental conditions of recent decades. Their prevalence has been steadily increasing, according to the most recent data from the Centers for Disease Control and Prevention (CDC, 2025); ASD currently affects approximately 32.2 per 1000 children, corresponding to about 1 in every 31 eight-year-olds [1]. This rising incidence has major implications not only for affected individuals and their families but also for healthcare systems and society as a whole.

The etiology of ASD is now understood as the outcome of a multifactorial interaction between genetic, epigenetic, and environmental influences [2]. While genetic factors have been extensively characterized, with hundreds of gene variants identified as conferring increased susceptibility, the increase in incidence and its enormous clinical variability cannot be explained by genetics alone [3,4,5]. Consequently, growing attention has turned toward early environmental factors that can shape brain development from conception through early childhood.

Within this context, the gut microbiota has emerged as one of the most promising and intriguing biological modulators. This vast ecosystem, composed of trillions of microorganisms including bacteria, viruses, and fungi, maintains a dynamic symbiotic relationship with the human host and exerts essential roles in metabolism, immune regulation, and neural function [6,7]. In recent years, evidence has expanded linking the gut microbiota not only to gastrointestinal health but also to brain development and the regulation of social, emotional, and cognitive behaviors through the so-called microbiota–gut–brain axis (MGBA) [8,9,10]. These discoveries have opened novel avenues for exploring the biological underpinnings of neuropsychiatric conditions, including ASD [11,12].

Interest in the microbiota–ASD connection has been driven by several key observations:The high prevalence of gastrointestinal symptoms among children with ASD exceeds that observed in typically developing peers [13,14,15,16].The recurrent patterns of gut dysbiosis identified in metagenomic studies [17,18,19].Preclinical findings demonstrate that alterations in microbial composition can modulate social and anxiety-like behaviors in animal models [20].Early clinical evidence indicates that nutritional or microbiota-targeted interventions may influence behavioral outcomes [21,22,23,24].

The present review aims to provide an updated and integrative overview of the relationship between the gut microbiota and autism. Specifically, it will explore the biological mechanisms underlying gut–brain communication, the impact of maternal microbiota during pregnancy and lactation on early neurodevelopmental and epigenetic programming, and the current clinical evidence for nutritional and therapeutic strategies targeting the microbiota.

This work does not propose the microbiota as the sole explanatory key to ASD, but rather as a crucial regulatory element within a broader network of genetic, environmental, and immunological interactions that collectively shape neurobehavioral development.

## 2. Materials and Methods

A comprehensive literature search was conducted using PubMed, Scopus, and Web of Science databases to identify relevant studies published between 2010 and 2025. Search terms included “autism spectrum disorder”, “gut microbiota”, “microbiota–gut–brain axis”, “maternal microbiota”, “nutrition”, and “neurodevelopment”. Boolean operators (AND, OR) were applied to refine search results, and reference lists of key papers were manually screened to identify additional studies.

We included original research articles, systematic reviews, and meta-analyses published in English, focusing on the interaction between the gut microbiota, neurodevelopment, and nutritional or microbiota-targeted interventions in ASD. Case reports, letters, and conference abstracts were excluded. Both human and animal studies were reviewed to provide an integrative understanding of the topic.

Three reviewers independently screened titles and abstracts; full texts of potentially eligible studies were assessed for inclusion. Disagreements were resolved through discussion with a fourth reviewer.

A total of 180 records were initially retrieved (110 from PubMed, 40 from Scopus, and 30 from Web of Science). After removing duplicates and non-relevant papers, 143 studies met the inclusion criteria and were analyzed. Results were synthesized and organized into thematic areas describing the microbiota–gut–brain axis, immune–inflammatory pathways, maternal microbiota, and dietary influences on ASD (Figure 1).

## 3. Results and Discussion

### 3.1. Microbiota–Gut–Brain Axis, Biological Barriers, and Neurodevelopment

The term microbiota–gut–brain axis (MGBA) does not describe an anatomical axis, but a pathophysiological construct that integrates immune, neuroendocrine, metabolic, and neural pathways. This bidirectional network allows the microbiota to modulate the permeability of biological barriers, microglial activation, stress response, and the production of neuroactive metabolites. In the context of autism, alterations in these mechanisms may contribute to neuroinflammation, impaired microglial maturation, and changes in the synthesis of serotonin, short-chain fatty acids (SCFAs), and tryptophan derivatives, processes now recognized as relevant to the pathophysiology of ASD. Although the gastrointestinal tract and central nervous system (CNS) are anatomically distant, well-established evidence demonstrates their bidirectional communication via the gut microbiota. The expansion of sequencing technologies and data from the Human Microbiome Project has expanded the understanding of the complexity and functional role of the human microbiome. The MGBA network integrates neuroendocrine, immune, metabolic, and neural pathways, including the hypothalamic–pituitary–adrenal (HPA) axis, the enteric nervous system (ENS), the vagus nerve, the immune system, and central and peripheral biological barriers [25,26]. Through these integrated mechanisms, the gut microbiota influences central processes such as peripheral and central barrier permeability, immune homeostasis, neurotransmission, functional CNS homeostasis, and stress response, effectively participating in neurodevelopment [27,28,29].

### 3.2. Intestinal Immune Barrier and Blood–Brain Barrier

The intestinal epithelium represents the largest interface between the external environment and the host, with an area of about 400 m^2^. This vast surface area is responsible for the digestion and absorption of nutrients, simultaneously acting as a selective barrier that tolerates commensal microorganisms and food antigens but excludes pathogens and toxins. The intestinal immune barrier (GIB) is organized into three main layers: a mucus layer, which houses the mucobiota, microorganisms that reside near the surface of the mucosa and release bioactive products that modulate host immunity [29,30], an epithelial layer, consisting of enterocytes, goblet cells, Paneth cells, M cells and enteroendocrine cells interconnected by tight junctions (e.g., occludin, claudine, ZO-1, JAM), which ensure the integrity of the barrier, and the lamina propria, which contains the lymphoid tissue associated with the intestine (GALT) populated by innate immune cells (mast cells, dendritic cells, macrophages) and adaptive immune cells. Communication between epithelial and immune cells is mediated by a complex network of signaling molecules that maintains homeostasis and prevents microbial translocation. When this balance is disrupted, due to mucus thinning, epithelial damage, or tight junction impairment, microbes and bacterial components such as lipopolysaccharides (LPSs) can penetrate the lamina propria, triggering pro-inflammatory responses [31,32,33].

The BBB serves as a highly selective second defense system within the CNS. Unlike peripheral vascularization, the endothelial cells of the BBB are tightly united, restricting the passage of most molecules to preserve neural integrity. Violations of BBB function are a hallmark of neuroinflammatory and neurodegenerative disorders. In addition to the BBB, the CNS possesses the choroid plexus vascular barrier (PVB), which separates the bloodstream from the cerebrospinal fluid (CSF) [34]. PVB is more permeable than the BBB due to the presence of fenestrated capillaries, but this permeability is regulated by the transmembrane protein PV-1 (plasmalemmal vesicle-associated protein-1). Under physiological conditions, gut eubiosis contributes to the normal functioning of these vascular barriers, ensuring that only beneficial bioactives reach the brain. Conversely, gut dysbiosis and systemic inflammation can compromise the integrity of the BBB. In these cases, PVB reduces its permeability to block the entry of harmful molecules such as LPS into the CNS. However, this defense mechanism also limits the exchange of essential nutrients and impairs the elimination of neurotoxic waste [35].

Together, the intestinal and brain barriers form a coordinated “shield” that safeguards the body’s overall homeostasis. Eubiosis contributes to the proper functioning of both barriers, while dysbiosis can cause selective reduction in permeability and accumulation of neurotoxic metabolites.

### 3.3. Endocrine, Neural, and Metabolic Pathways in Microbiota–Gut–Brain Communication

The endocrine system, in particular the HPA axis, represents a key regulatory pathway within the MGBA. The HPA axis modulates stress responses through the release of corticotropin-releasing hormone (CRH), adrenocorticotropic hormone (ACTH), and cortisol. Experimental evidence indicates that germ-free (GF) mice exposed to mild stress show an exaggerated increase in ACTH and corticosterone levels compared to conventionally colonized animals, underscoring the role of the microbiota in calibrating neuroendocrine stress responses [36]. HPA axis dysregulation has long been associated with psychiatric and neurodevelopmental conditions, including anxiety, depression, and autism. During stress, the paraventricular nucleus (PVN) of the hypothalamus secretes CRH, stimulating the release of ACTH from the anterior pituitary gland, which in turn induces cortisol synthesis by the adrenal cortex. Cortisol exerts feedback control through glucocorticoid and mineralocorticoid receptors in the brain, affecting emotional regulation and cognition. Maternal stress, environmental toxins, and nutritional deficiencies can hyperactivate the HPA axis, leading to maternal dysbiosis and increased risk of neurodevelopmental alterations in offspring. Studies in animal models have shown that exposure to LPS during puberty induces persistent depressive-like behavior in adulthood. Collectively, these findings suggest that both prenatal and postnatal stressors modulate microbiota composition and HPA axis activity, potentially contributing to the pathophysiology of ASD and related disorders. While current evidence primarily supports correlation rather than causation, mechanistic links between microbial ecology, stress, and neurodevelopment remain an active area of investigation [37,38,39,40].

### 3.4. Neural Pathways

Neural communication between the gut and the brain is primarily mediated by the enteric nervous system (ENS) and the vagus nerve. The vagus nerve provides direct bidirectional signaling between the gut and the CNS. It regulates visceral functions such as heart rate, digestion, and endocrine activity, and modulates the release of gut neurohormones, including serotonin, peptide YY (PYY), cholecystokinin (CCK), and glucagon-like peptide-1 (GLP-1). Experimental vagotomy models reveal that disruption of vagal signaling impairs cognition, neurogenesis, and stress responsiveness, while vagus nerve stimulation promotes hippocampal neurogenesis and improves levels of brain-derived neurotrophic factor (BDNF), key to synaptic plasticity and neural repair [41]. These findings highlight that the vagus nerve is a crucial conduit for microbiota-derived signals to influence central brain processes. The close coupling between ENS and CNS explains why microbiota-targeted therapies, such as fecal microbiota transplantation (FMT), have shown promising effects not only on gastrointestinal symptoms, but also on behavioral manifestations in ASD and other neuropsychiatric disorders [42,43].

### 3.5. Microbial Metabolites and Neurotransmitters

Numerous microbial metabolites act as signaling molecules within the MGBA. Among these, serotonin (5-HT) is particularly remarkable: about 95% of total serotonin is synthesized in the gut by enterochromaffin cells, with only 5% produced by central serotonergic neurons [44]. Dietary tryptophan is converted to 5-hydroxytryptophan and then to serotonin, affecting both intestinal motility and central mood regulation. Tryptophan metabolism also follows the kynurenine pathway, generating neuroactive compounds such as kynurenic acid (neuroprotective) and quinolinic acid (neurotoxic). Under conditions of stress or inflammation, excessive activity of tryptophan-2,3-dioxygenase (TDO) or indoleamine-2,3-dioxygenase (IDO) distorts metabolism to neurotoxic products, reducing serotonin bioavailability and contributing to neural dysfunction [45]. Other microbiota-derived neurotransmitters include dopamine and γ-aminobutyric acid (GABA). Several lactic acid bacteria, including bifidobacteria, express glutamate decarboxylase (GAD), which converts dietary glutamate into GABA, a key inhibitory neurotransmitter that can affect stress responses and pain modulation through vagal pathways [46]. SCFAs such as acetate, propionate, and butyrate also mediate gut–brain communication. Acting through G-protein-coupled receptors (FFAR2, FFAR3, HCAR2), SCFAs regulate epithelial integrity, immune homeostasis, and neurotransmitter synthesis. Butyrate, in particular, inhibits histone deacetylase (HDAC), thereby influencing epigenetic regulation and microglial activation, promoting neuroprotection. Finally, bile acids (BAs), derived from cholesterol metabolism, exert similar effects through receptors such as TGR5 and FXR on microglial cells. Altered BA profiles have been linked to impaired cognitive and neurodegenerative disorders [47,48].

In summary, numerous microbial metabolites orchestrate MGBA communication through multiple mechanisms, from neurotransmitter synthesis to epigenetic and immune modulation. These complex pathways highlight the interconnection between diet, microbiota, and neurological function.

### 3.6. Microglia, Inflammation, and Neurocognitive Development

Microglia, immune cells resident in the CNS, play a fundamental role in neurological development, synaptic pruning, and in the defense and functional homeostasis of the brain. Originating from the erythromyoid progenitors of the yolk sac, microglia migrate to the neural tube during embryogenesis and differentiate under the influence of factors such as TGF-β and IL-34. Under physiological conditions, microglia oscillate between two functional states: a branched resting phenotype (M2) responsible for surveillance and homeostasis, and an amoeboid activated phenotype (M1) involved in phagocytosis and immune defense [49]. After exposure to microbial products such as LPS, microglia polarize to the pro-inflammatory M1 state, producing cytokines (TNF-α, IL-6) and reactive oxygen species (ROS) [50]. Normally, this activation is transient and followed by IL-4 and IL-13-mediated resolution, restoring M2 homeostasis. However, chronic gut dysbiosis and increased intestinal permeability may support persistent activation of M1 microglia, leading to persistent neuroinflammation [51]. Chronic systemic inflammation (CSI) is increasingly recognized as a factor in multi-organ dysfunction, metabolic disorders, and neurodevelopmental conditions, including ASD. During pregnancy, maternal immune activation (MIA) represents a critical inflammatory event that can induce epigenetic changes in the developing fetal brain, potentially affecting neurobehavioral outcomes later in life [52,53,54].

### 3.7. Direct Implications for ASD

The increase in the prevalence of ASD is associated with environmental and perinatal factors related to MGBA modulation: maternal microbiota imbalance during pregnancy, prematurity, birth by cesarean section, use of infant formula, industrial diet, high ω6/ω3 ratio, and perinatal antibiotics (Figure 2) [55,56,57]. Although they do not prove causation, such associations indicate that the early environment critically influences microbial colonization trajectory and immune and neural maturation [58,59].

Children with ASD show a reduced abundance of beneficial taxa (*Bifidobacterium*, *Prevotella*) and increased potential pathobionts (*Clostridium* spp.), patterns similar to those seen in infants born by cesarean section [60,61]. The latter is associated with a 26–33% increased risk of ASD or attention-deficit hyperactivity disorder (ADHD) [62]. Maternal obesity (BMI ≥ 30 kg/m^2^) during pregnancy also increases the risk of ASD in offspring by 1.5 to 2 times through dysbiosis, systemic inflammation, metabolomic alterations, and alterations in fetal epigenetic programming [63]. These converging factors appear to divert microbial and neural development from their physiological trajectory. The resulting imbalance favors a pro-inflammatory state characterized by microglial polarization toward the M1 phenotype, perpetuating neuroinflammation and impairing synaptic pruning and plasticity [64]. Microbiota-derived metabolites, including SCFAs and tryptophan compounds, modulate microglial activity. Excess propionate, associated with *Bacteroides* species, has been linked to hyperexcitability and repetitive behaviors [65].

### 3.8. Clinical Evidence: Taxonomy, Metabolomics, Immunology, and Extraintestinal Microbiota

#### 3.8.1. Taxonomic Evidence

High-throughput sequencing technologies, including 16S rRNA gene profiling and shotgun metagenomics, have revealed significant differences in gut microbiota between children with ASD and neurotypical controls. Some studies even suggest that the abundance of specific taxa may be related to the severity of symptoms. However, variability in diet, medication use, lifestyle, and comorbidities has important confounders, complicating the identification of a consistent microbial signature for diagnostic or predictive purposes [66,67,68].

Although data on microbial diversity remains inconsistent, some patterns have emerged across all studies:Reduced abundance of beneficial taxa such as *Bifidobacterium*, *Prevotella*, and butyrate-producing families (*Faecalibacteriaceae*, *Lachnospiraceae*, *Ruminococcaceae*) [69];Increased abundance of potentially pro-inflammatory and neuroactive species, including *Clostridium* spp., *Desulfovibrio*, and *Veillonella* [70].These taxa affect the production of key metabolites such as SCFAs, neurotransmitter precursors, and inflammatory mediators, thereby affecting brain function and behavior [71].

#### 3.8.2. Metabolic Evidence

ASD is associated with a reduction in SCFAs produced by *Ruminococcaceae*, *Lachnospiraceae*, *Eubacterium rectale* and *Erysipelotrichaceae* [72]. *Prevotella*, *Bifidobacterium* and *Ruminococcus* produce acetate; *Lachnospiraceae* and *Ruminococcaceae* generate butyrate; *Bacteroides*, *Prevotella*, *Rosobria*, and *Blautia* produce propionate [73].

The reduction in *Bifidobacterium* also impairs the synthesis of folate and B12, key cofactors in methyl metabolism, neurotransmitter synthesis and DNA methylation. Low plasma levels of folate and B12, frequently observed in ASD, are further aggravated by selective dietary patterns characterized by high sugar content and low fruit and vegetable intake. Tryptophan metabolism is also altered, with reduced serotonin and alterations in social behavior [74,75,76,77].

In summary, in individuals with ASD, microbiota alterations reduce the production of SCFAs and the availability of key nutrients such as folate and B12, affecting metabolism, neurotransmitter synthesis, and epigenetic regulation. These metabolic changes correlate with changes in behavioral and neurological domains.

#### 3.8.3. Immune–Inflammatory Tests

Numerous studies show elevated levels of TNF-α, IFN-γ and pro-inflammatory immune profiles associated with increased clinical severity in ASD. Dysbiosis is linked to increased intestinal permeability, microglial activation, and systemic inflammation [78,79,80].

#### 3.8.4. Extraintestinal Microbiota

Emerging evidence indicates that microbial communities outside the gut, particularly the oral microbiota, can also influence neurobehavioral outcomes. In a large cohort of more than 2000 U.S. families, Manghi et al. analyzed 7000 saliva metagenomes and identified distinct oral microbial signatures associated with ASD. These differences were particularly evident in metabolic pathways related to the degradation of serotonin, GABA, and dopamine. Although the direct causal links remain speculative, such findings suggest that oral microorganisms capable of metabolizing neurotransmitters could indirectly affect the gut–brain axis. The field is still in its infancy, but these data open new possibilities for exploring non-invasive biomarkers for ASD [81,82].

#### 3.8.5. Study Limitations

Despite substantial progress, research on the microbiota–ASD relationship faces significant methodological challenges. Many studies rely on small sample sizes, cross-sectional designs, and heterogeneous populations that differ in diet, medications, and comorbidities. In addition, methodological inconsistencies in the collection, storage, and sequencing of fecal samples hinder comparability. These limitations make it difficult to determine whether dysbiosis is a causal factor in the pathogenesis of ASD or a secondary consequence of an altered lifestyle and diet. At present, no universal microbial biomarker or diagnostic signature for ASD has been validated, underscoring the need for standardized, longitudinal, and mechanistic studies in different populations.

## 4. Diet, Microbiota, and Autism: Between Empiricism and Personalization

In early life, the gut microbiota profoundly influences the maturation of the immune system; approximately 70–80% of immune cells reside in the gut-associated lymphoid tissue (GALT) [29]. Therefore, appropriate nutrition and targeted dietary interventions during pregnancy and early childhood are critical to support microbial diversity, immune homeostasis, and optimal neurodevelopmental outcomes.

### 4.1. The Western Diet and Its Consequences

Over the past five decades, the Western diet (WD) has become the dominant dietary pattern in industrialized societies. It is characterized by low intake of fruits, vegetables, legumes, and whole grains, resulting in poor fiber and bioactive compound content, and a high consumption of refined carbohydrates and ultra-processed foods (UPFs) [83,84,85]. These foods are typically rich in sugars, saturated fats, salt, and artificial additives such as emulsifiers, colorants, and preservatives, while being deficient in essential micronutrients (polyphenols, omega-3 fatty acids, vitamins B6, B9, B12, C, D, and E, and trace elements). Animal studies demonstrate that dietary emulsifiers, such as polysorbate 80 (P80) and carboxymethylcellulose (CMC), disrupt intestinal barrier integrity, decrease microbial diversity, and promote systemic inflammation by facilitating bacterial translocation into the lamina propria [86,87,88,89]. A systematic review encompassing over 260,000 participants revealed a linear relationship between UPF intake and depression risk [90]. Another review of 39 meta-analyses corroborated these findings, linking UPF consumption with a higher incidence of depression and common mental disorders [91]. Moreover, a WD rich in sugars and refined flours leads to glycation and oxidation end-products (AGEs and ALEs) during high-temperature cooking, increasing oxidative stress and promoting inflammation [92]. Excessive salt intake, especially from non-iodized, prepackaged foods, has also been associated with decreased *Lactobacillus* abundance, compromising intestinal and immune balance [93,94,95].

Collectively, these dietary features foster dysbiosis, barrier permeability, systemic inflammation, and metabolic dysfunction, all of which are recognized as potential contributors to neuroinflammatory and neurodevelopmental disorders, including ASD.

### 4.2. Exclusion Diets: Evidence and Controversies

In recent years, various exclusion diets have gained popularity among families of children with ASD. These include gluten-free (GFD), casein-free (CFD), gluten- and casein-free (GFCF), anti-candida, Feingold, and Gut and Psychology Syndrome (GAPS) diets. While often adopted empirically or based on anecdotal reports, the scientific evidence supporting their efficacy remains limited and inconsistent. The anti-candida diet restricts refined carbohydrates, sugars, red meat, and dairy products. The GAPS diet eliminates starchy vegetables, grains, and sweeteners while emphasizing meat, fish, and fermented foods. The Feingold diet excludes salicylates, preservatives, and certain fruits such as grapes, apples, and peaches. The GFCF diet removes gluten and casein, favoring fish, vegetables, and lean proteins. A meta-analysis found that the GFCF diet did not produce significant improvements in social, behavioral, or cognitive outcomes in children with ASD, and in some cases worsened gastrointestinal symptoms [96]. Similarly, the European Society for Pediatric Gastroenterology, Hepatology and Nutrition (ESPGHAN) recommends GFD or CFD only for patients with diagnosed celiac disease or proven intolerance [97]. Unsupervised restrictive diets can lead to nutrient deficiencies, including low levels of calcium, vitamin D, folate, and essential fatty acids [98]. Moreover, such diets often reduce microbial diversity and increase reliance on ultra-processed “free-from” products high in sugar and additives, further aggravating metabolic imbalance.

Therefore, these dietary patterns are not supported for general use in ASD management. Large-scale, randomized, longitudinal trials are needed to clarify whether specific subgroups may benefit from tailored dietary interventions.

## 5. Emerging Therapeutic Diets

In contrast to exclusionary approaches, several structured dietary interventions show potential for microbiota modulation and neurobehavioral benefits.

### 5.1. Ketogenic Diet (KD)

The ketogenic diet (KD) is a high-fat, low-carbohydrate, normoproteic regimen that induces ketosis. Initially developed for refractory epilepsy, KD has demonstrated potential benefits in neurodegenerative and neurodevelopmental disorders, including ASD [99,100,101]. Ketone bodies (KBs) act as alternative energy substrates for neurons and glial cells and exert neuroprotective and anti-inflammatory effects. Mechanistically, KD enhances mitochondrial function, reduces oxidative stress, and inhibits pro-inflammatory signaling pathways such as NF-κB, TNF-α, IL-2, and IL-6 [102,103]. Furthermore, it promotes microglial polarization toward the M2 (anti-inflammatory) phenotype and upregulates BDNF expression. Animal studies also suggest that KD reshapes gut microbiota composition by reducing *Actinobacteria*, *Akkermansia muciniphila*, *Bacteroides fragilis*, and *Bilophila wadsworthia*, thereby restoring metabolic homeostasis [101]. However, it is important to emphasize that KD requires close medical and nutritionist monitoring and supervision, with adequate supplementation and customization to avoid nutritional deficiencies.

### 5.2. Low-Glycemic-Index and Low-FODMAP Diets

The low-glycemic-index diet (LGID) emphasizes foods that cause gradual increases in blood glucose, including whole grains, legumes, and fibrous vegetables. In animal models, LGID has been associated with decreased neuroinflammation and oxidative stress [104].

The low-FODMAP diet, designed to reduce fermentable oligosaccharides, disaccharides, monosaccharides, and polyols, can alleviate gastrointestinal symptoms such as bloating and diarrhea frequently reported in ASD [105,106,107]. Although preliminary results are promising, evidence remains limited, and long-term studies are required to assess safety and efficacy in pediatric populations. However, these dietary regimens cannot be improvised through “do-it-yourself” methods but require the monitoring of specialized doctors and nutritionists.

### 5.3. Food Selectivity and Nutritional Deficiencies

Food selectivity, defined as restricted food variety combined with sensory sensitivities to taste, texture, color, or temperature, affects 46–89% of children with ASD [108,109,110]. This pattern is often associated with nutrient deficiencies, including low intake of omega-3 fatty acids, vitamins A, B6, B12, C, and folate, as well as minerals such as zinc, calcium, magnesium, and iron. These deficiencies may impair cognitive development, immune competence, and metabolic regulation [111,112].

Additionally, highly selective eating behaviors contribute to social and familial stress, further complicating dietary management. Restrictive nutritional interventions in such children must therefore be approached cautiously and always under multidisciplinary supervision, including pediatricians, nutritionists, and behavioral therapists.

## 6. The Need for Personalized Nutrition

Given the heterogeneity of ASD and the interindividual variability of microbiota composition, a “one-size-fits-all” dietary approach is neither realistic nor advisable. Nutritional strategies should be personalized, taking into account age, sex, genetic background, metabolic profile, comorbidities, and cultural dietary patterns.

Furthermore, because ASD diagnosis typically occurs within the first three years of life, maternal and early-life nutrition are particularly critical. Prospective cohort studies in mother–child pairs are essential to explore the role of maternal diet and microbiota during pregnancy and lactation in shaping neurodevelopmental outcomes. Early intervention through maternal nutritional optimization may represent one of the most promising strategies for reducing ASD risk and improving long-term developmental trajectories.

### 6.1. Maternal Diet, Functional Foods, and the Environment

Maternal nutrition and gut microbiota composition during pregnancy play pivotal roles in shaping fetal neurodevelopment. The intrauterine environment provides the foundation for the infant’s immune, metabolic, and neurological systems [8,113]. Numerous studies have demonstrated that maternal diet, body composition, and microbial balance directly influence epigenetic programming and brain development, suggesting that pregnancy represents a critical window for intervention [114,115].

During pregnancy, profound hormonal, metabolic, and immune changes alter the composition of the maternal gut microbiota. These changes are thought to improve energy absorption and modulate immune tolerance to support fetal growth. However, maternal dysbiosis, triggered by factors such as obesity, a high-fat diet, exposure to antibiotics, exposure to toxins and environmental pollutants, or chronic stress, has been associated with an increased risk of ASD, systemic inflammation, and neurodevelopmental disorders in offspring. Exposure to environmental pollutants, pesticides, heavy metals, and endocrine-disrupting chemicals can influence the composition of the microbiota and maternal-fetal immunometabolic processes.

Once again, much of the evidence is based on experimental or observational studies and suggests associations, not causation. Therefore, further research is needed to determine whether and how these exposures contribute to the risk of ASD [116,117,118,119,120]. Animal studies demonstrate that MIA can disrupt microglial maturation and alter fetal brain architecture through cytokine-mediated pathways (e.g., IL-6, IL-17A, TNF-α). In humans, elevated maternal inflammatory markers during gestation have been linked to a higher incidence of ASD and cognitive impairment in children [121,122,123,124]. Moreover, maternal dysbiosis affects microbial metabolites that cross the placental barrier, including SCFAs, bile acids, and tryptophan derivatives. These metabolites influence gene expression and synaptic development in the fetal brain via epigenetic regulation (DNA methylation, histone acetylation) [125]. Hence, maintaining a balanced maternal microbiota may be essential for optimal neurodevelopmental programming.

### 6.2. Functional Foods and Bioactive Nutrients

Functional foods and bioactive nutrients capable of modulating the microbiota are receiving growing attention for their potential role in preventing ASD and other neurodevelopmental disorders. Long-chain polyunsaturated fatty acids (PUFAs), particularly docosahexaenoic acid (DHA) and eicosapentaenoic acid (EPA), are critical for fetal brain development, neuronal membrane fluidity, and synaptogenesis [126,127]. Maternal omega-3 deficiency has been linked to increased risk of neuroinflammation, impaired cognition, and behavioral dysregulation. Supplementation during pregnancy has been shown to promote eubiosis by increasing *Bifidobacterium* and *Lactobacillus* abundance while reducing pro-inflammatory species such as Enterobacteriaceae [128]. Despite promising preclinical data, clinical trials report mixed outcomes regarding omega-3 supplementation and ASD prevention, indicating the need for better stratification and dosage standardization. Moreover, polyphenols, plant-derived compounds abundant in fruits, vegetables, tea, cocoa, and olive oil, exert potent antioxidant, anti-inflammatory, and microbiota-modulating properties. They enhance *Akkermansia muciniphila* and *Faecalibacterium prausnitzii*, both associated with intestinal barrier integrity and metabolic health. In animal models, polyphenol-rich diets improve cognitive performance and reduce oxidative stress in the developing brain [129]. Key dietary sources include flavonoids (berries, apples, onions), phenolic acids (coffee, whole grains), stilbenes (red grapes, peanuts), and lignans (seeds, legumes). However, bioavailability varies widely among individuals due to differences in gut microbiota composition, highlighting once again the importance of personalized nutrition.

Moreover, the use of probiotics, defined by live microorganisms that confer health benefits to the host, has emerged as a promising adjunct for maternal and neonatal health. Strains such as *Lactobacillus rhamnosus GG*, *Bifidobacterium longum*, and *Lactobacillus casei Shirota* have demonstrated immunomodulatory, anti-inflammatory, and gut-barrier-protective properties [130,131,132]. Probiotic supplementation during pregnancy and lactation has been shown to reduce the incidence of gestational diabetes, postpartum depression, and infant eczema, suggesting broader systemic effects mediated by microbiota modulation. Therefore, prebiotics, including inulin, galactooligosaccharides (GOSs), and fructooligosaccharides (FOSs), selectively stimulate beneficial bacterial growth and the production of SCFAs [133]. Maternal intake of prebiotics has been linked to enhanced microbial diversity in offspring, improved stress resilience, and reduced anxiety-like behavior in animal models [134] (Table 1). Although clinical evidence regarding probiotics, prebiotics, and ASD prevention remains preliminary, the biological plausibility is strong, and several clinical trials are underway to evaluate maternal and infant supplementation strategies.

## 7. Environmental Factors and Epigenetic Modulation

Beyond diet, multiple environmental exposures can influence both maternal microbiota and neurodevelopmental outcomes. Key contributors include pesticides and heavy metals (e.g., organophosphates, mercury, lead), which disrupt microbial composition and interfere with mitochondrial and synaptic function; endocrine-disrupting chemicals (EDCs) such as bisphenol A (BPA) and phthalates, which mimic hormonal activity and alter gut microbial metabolism [139,140,141,142]; and air pollution, associated with increased oxidative stress and neuroinflammation; perinatal antibiotic use reduces microbial diversity during critical developmental windows [143,144].

These exposures may modify epigenetic processes, including DNA methylation, histone modification, and microRNA expression, thereby affecting gene regulation in the developing brain [19]. The interplay between maternal diet, microbiota, and environmental stressors thus represents a multidimensional system that collectively shapes neurodevelopmental outcomes.

## 8. Toward Preventive and Translational Approaches

Understanding the maternal microbiota’s influence on fetal brain development opens new perspectives for preventive medicine. Maternal dietary optimization, microbial modulation, and lifestyle interventions could form the basis of personalized strategies to reduce ASD risk and enhance neurodevelopmental health.

Advancing this field will require research strategies that can capture the complexity of early-life biological development. One promising direction involves establishing longitudinal mother–infant cohorts that integrate metagenomics, metabolomics, epigenomics, and detailed clinical metadata. Such approaches would allow researchers to map microbiota-driven developmental pathways with far greater precision than is currently possible.

At the same time, controlled trials evaluating functional foods and probiotics are essential for determining not only their efficacy and safety, but also the most appropriate timing for supplementation during early development. These trials would provide the experimental evidence needed to move beyond associative findings.

Finally, broader public health initiatives will play a crucial role. Strategies that support balanced maternal nutrition, minimize exposure to environmental toxins, and encourage responsible antibiotic use could help create conditions that favor healthy microbiota development from the very beginning of life.

The maternal microbiota is not merely a passive ecosystem but an active biological interface that connects the environment, diet, and genetic background of the mother to the emerging physiology of the child. Protecting and nurturing this ecosystem may represent one of the most promising frontiers in the prevention of neurodevelopmental disorders.

## 9. Current ASD and Microbiota Limits and Future Prospects

Although a growing number of studies describe associations between alterations in the gut microbiota and behavioral or clinical characteristics of ASD, the causal link between dysbiosis and neurodevelopment remains largely undetermined. In fact, much of the available evidence derives from observational, cross-sectional, or small-sample studies, which do not allow for establishing the causal directionality of the phenomena or for excluding the central role of confounding factors. In many cases, it is unclear whether microbiotic differences represent an etiological mechanism, a consequence of the ASD phenotype (e.g., due to food selectivity, gastrointestinal comorbidities, or drug use), or a co-factor modulated by genetics, epigenetics, environment, lifestyle, and diet.

### 9.1. Methodological Limitations of the Available Studies

The current literature faces several methodological challenges that make it difficult to establish a causal link between gut microbiota alterations and ASD. A major issue is that most studies rely on cross-sectional designs, comparing children with ASD and neurotypical controls at a single point in time. This approach does not allow researchers to determine whether dysbiosis emerges before the onset of symptoms or develops as a consequence of them.

Another obstacle comes from the considerable heterogeneity across studies. Differences in sample collection procedures and sequencing platforms often produce results that are difficult to compare, limiting the consistency and reproducibility of findings. Adding to this complexity, many studies do not adequately control for important confounding factors such as diet, medication use, or comorbidities, all of which are known to influence microbial composition.

Finally, the marked phenotypic heterogeneity of ASD itself, coupled with the frequent lack of stratification within study samples, can obscure meaningful biological patterns. Without distinguishing among different ASD presentations, relevant associations between microbial profiles and specific clinical features may remain hidden.

Therefore, the often-conflicting results (in terms of taxonomic and metabolomic profiles) of the various studies need critical interpretation. Finally, animal models certainly provide important mechanistic clues, but they are not directly generalizable and transferable to human physiology.

### 9.2. Future Prospects

To establish a true causal relationship, future research will need to adopt more robust and comprehensive study designs. This includes the development of large longitudinal cohorts that follow children from pregnancy through early childhood, allowing researchers to observe how microbiota changes unfold over time. Equally important is the use of integrated multi-omics approaches, which can combine genomic, metabolomic, and other molecular data to provide a more complete view of host–microbiota interactions.

Causality could also be better assessed through carefully controlled interventions that directly test how specific modifications of the microbiota influence developmental outcomes. Finally, studies will need to incorporate rigorous biological stratification, identifying subgroups of participants who are homogeneous from a biological perspective. Such stratification would help uncover patterns that may be obscured when highly heterogeneous groups are analyzed together.

In this regard, the GEMMA (Genome, Environment, Microbiome and Metabolome in Autism) study, Troisi J et al. [66], represents an advanced methodological model and among the most rigorous, based on a longitudinal follow-up, with integration of metagenomics, metabolomics, epigenomics, and immune analyses, and rigorous evaluation of environmental and nutritional factors. This approach could make it possible to identify early biomarkers, clarify the timeline of events, and distinguish cause, consequence, and co-factors.

## 10. Conclusions

Understanding the complex interactions between the gut microbiota and ASD represents one of the most fascinating and promising challenges in biomedical research today. Emerging evidence indicates that the microbiota is not a mere bystander, but an active player in modulating neurobiological, behavioral, and immune processes that can contribute to neurodevelopment. However, research cannot remain confined to laboratories or scientific publications: it must be translated into concrete public health tools. Prevention is a political and ethical act because it means investing in the health of current and future generations. It is therefore necessary to transform current knowledge about the microbiota and neurodevelopment into clinical, social, and health actions that offer real solutions to people with ASD and their families.

From this perspective, focusing on prevention, promoting healthy lifestyles, and integrating new technologies represent the true tools for building a more sustainable and inclusive healthcare system. Only through an interdisciplinary approach will it be possible to transform microbiota research from a scientific promise into the foundation of a new global health paradigm.

## Figures and Tables

**Figure 1 nutrients-17-03706-f001:**
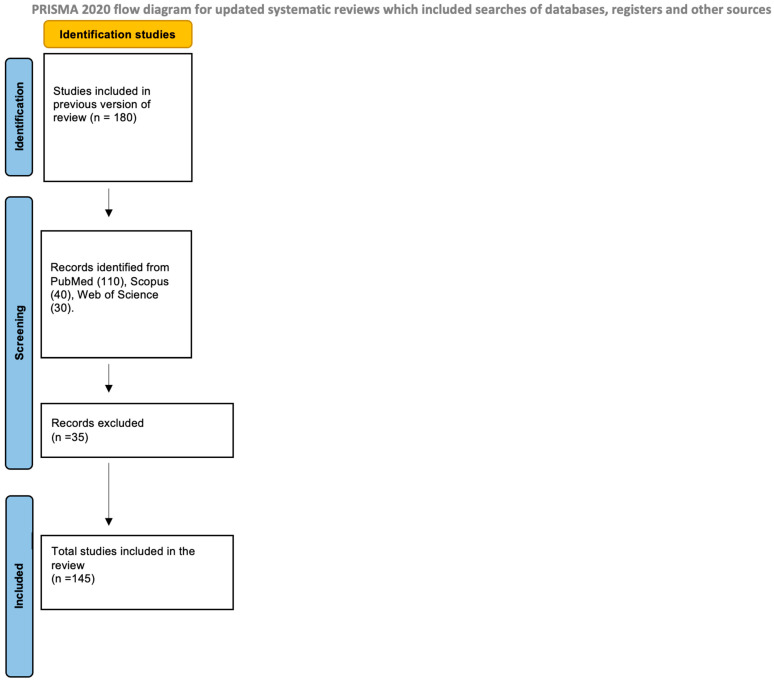
PRISMA flow diagram.

**Figure 2 nutrients-17-03706-f002:**
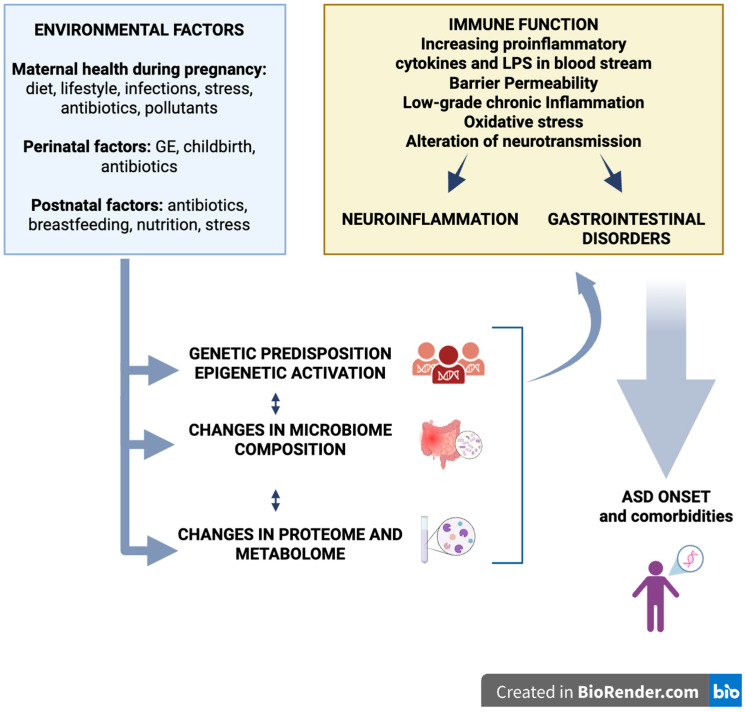
Genetics and epigenetic factors in ASD onset and comorbidities.

**Table 1 nutrients-17-03706-t001:** Key microbial changes in ASD and results of microbiota interventions.

Intervention	Main Effects on the Microbiota	Reported Clinical/Metabolic Outcomes
Western Diet (WD)	↓ *Bifidobacterium*, ↓ *Prevotella*, ↓ *butyrate-producing*; ↑ *Clostridium* spp., ↑ *Desulfovibrio*	↑ intestinal permeability, ↑ inflammation; (preclinical) [135]
Diet rich in fiber, vegetables, polyphenols	↑ *Akkermansia*, ↑ *F. prausnitzii*, ↑ *SCFAs*, ↑ *Bifidobacterium*	Improved gut barrier, ↓ inflammation, cognitive benefits [132]
Fermented foods	↑ *Lactobacillus*, ↑ *Bifidobacterium*, ↑ *diversity*	Reduction in inflammation, improvement of GI symptoms [21]
Exclusion diets (GFCF, other)	*Variable effects*; *possible* ↓ *diversity*; ↓ *Bifidobacterium*	Inconsistent evidence; Risk of nutritional deficiencies [98]
Low-FODMAP	↓ *Prevotella*, ↓ *Bacteroides*	Improvement of GI symptoms; Inconclusive behavioral impact [107]
Low-GI diet	↑ *Anti-inflammatory metabolites*; ↑ *Diversity (preclinical)*	↓ oxidative stress and neuroinflammation (animal)
Ketogenic diet (KD)	↓ *Actinobacteria*, *Akkermansia*, *B. fragilis*, *Bilophila*	↓ inflammatory cytokines, ↑ BDNF; Possible behavioral benefits [101]
Prebiotics (inulin, GOS, FOS)	↑ *Bifidobacterium*, ↑ *SCFAs*, ↑ *Lactobacillus*	Barrier improvement, ↓ inflammation [132]
Probiotics	*Increased beneficial strains*; ↓ *Enterobacteriaceae*	GI improvement; Variable behavioral effects [136]
Postbiotics	↑ *beneficial metabolites* (e.g., *butyrate*)	Barrier improvement and immunomodulation (preclinical) [137]
FMT	↑ Diversity ↑ *Bifidobacterium*, ↑ *Prevotella*, ↓ *Desulfovibrio*	Persistent GI and behavioral improvement for up to 2 years [14,15,16,17,18,19,20,21,22,23,24,25,26,27,28,29,30,31,32,33,34,35,36,37,38,39,40,41,42,43,44,45,46,47,48,49,50,51,52,53,54,55,56,57,58,59,60,61,62,63,64,65,66,67,68,69,70,71,72,73,74,75,76,77,78,79,80,81,82,83,84,85,86,87,88,89,90,91,92,93,94,95,96,97,98,99,100,101,102,103,104,105,106,107,108,109,110,111,112,113,114,115,116,117,118,119,120,121,122,123,124,125,126,127,128,129,130,131,132,133,134,135,136,137,138]

↓ downregulation; ↑ upregulation.

## Data Availability

Data sharing is not applicable. No new data were created or analyzed in this study. Data sharing does not apply to this article.

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
