# Peer review of "Gut Microbiota and Autism: Unlocking Connections"

_nutrients, 2025, doi:10.3390/nu17233706_

Round 1
Reviewer 1 Report
Comments and Suggestions for Authors
The authors reviewed the connections between gut microbiota and ASD.
My biggest concern is the structure of the manuscript. There are "1. Introduction", "2. Materials & Methods", "3. Results", and "9. Toward Preventive and Translational Approaches", which I can understand. However, there are also "4. Discussion", "5. Diet, Microbiota, and Autism: Between Empiricism and Personalization", "6. Emerging Therapeutic Diets", "7. The Need for Personalized Nutrition", and "8. Environmental Factors and Epigenetic Modulation" that I cannot understand why the authors separated these topics. This is a review, not a research article; therefore, all parts after "Methods" should be organized as a narrative line.
And I doubt there are too many bullets, just like those generated through AI.
In my opinion, the authors have to reorganize the paragraphs in a logical way.
And there is not even one figure, table, or other illustration, which is really weird as a review.
There are many errors in the references. For instance, Ref. 27 and Ref. 28 are the same. Ref. 38, Ref. 39, and Ref. 41 are the same. I am afraid that the author did not treat this manuscript carefully and spend enough time on finishing it.
Author Response
Comments 1: My biggest concern is the structure of the manuscript. There are "1. Introduction", "2. Materials & Methods", "3. Results", and "9. Toward Preventive and Translational Approaches", which I can understand. However, there are also "4. Discussion", "5. Diet, Microbiota, and Autism: Between Empiricism and Personalization", "6. Emerging Therapeutic Diets", "7. The Need for Personalized Nutrition", and "8. Environmental Factors and Epigenetic Modulation" that I cannot understand why the authors separated these topics. This is a review, not a research article; therefore, all parts after "Methods" should be organized as a narrative line.
Response: We fully endorse the reviewer's comment regarding the organization of the paragraphs. We acknowledge that, given that this is a review, the inclusion of sections labeled "Discussion," "Diet, Microbiota and Autism," "Emerging Therapeutic Diets," "Need for Personalized Nutrition," and "Environmental Factors and Epigenetic Modulation" may seem fragmented rather than a more fluid narrative. However, given that the review is quite lengthy, we believe that dividing it into subchapters will aid the reader in a detailed understanding of the different focuses.
Comments 2: And I doubt there are too many bullets, just like those generated through AI. In my opinion, the authors have to reorganize the paragraphs in a logical way.
Response: Thank you for highlighting this point. We have revised the entire manuscript to drastically reduce the number of bullet points, replacing them with more conversational prose suited to the style of a narrative review.
Comments 3: And there is not even one figure, table, or other illustration, which is really weird as a review.
Response: We fully accept your observation. We've added, in addition to the graphic abstract, a figure and table that can help the reader get an overall picture.
Comments 4: There are many errors in the references. For instance, Ref. 27 and Ref. 28 are the same. Ref. 38, Ref. 39, and Ref. 41 are the same. I am afraid that the author did not treat this manuscript carefully and spend enough time on finishing it.
Response: Thank you for highlighting the duplications and errors in the citations. This was a bibliographical error that we have carefully corrected in the new version.
Reviewer 2 Report
Comments and Suggestions for Authors
The manuscript excellently integrates genetic, immunological, endocrine and neurometabolic perspectives. The cited sources reliably and relevantly support the authors’ claims. It is particularly valuable that the work emphasizes not only the biological mechanisms but also the environmental and lifestyle factors, as well as the potential role of nutritional interventions. The study is an outstandingly well-developed, in-depth, and coherent scientific review that effectively bridges microbiological, neurological, and clinical research domains.
Constructive questions: What strategies do the authors consider most promising for the clinical implementation of microbiome-based prevention, particularly in supporting maternal nutrition during pregnancy?
Have the authors considered examining the microbiome–ASD relationship within holistic models that integrate social or psychological factors (eg., stress, family dietary habits)?
Given the richness of the content, it may be helpful to include a few brief summary sentences or transitional paragraphs between sections to strengthen coherence and help readers follow the complex chains of biological processes more easily.
Author Response
Comment 1: Constructive questions: What strategies do the authors consider most promising for the clinical implementation of microbiome-based prevention, particularly in supporting maternal nutrition during pregnancy?
Response: We thank the reviewer for this question. In general, I could answer that the Mediterranean diet, rich in fiber, whole grains, legumes, and seeds, and revisited with the addition of fermented foods, is a useful eubiotic strategy for pregnant women. However, from a personalization perspective, it would be interesting to know the microbiota of the pregnant woman being cared for so that possible therapeutic support with pre-, pro-, and post-biotics can also be individualized.
Comments 2: Have the authors considered examining the microbiome–ASD relationship within holistic models that integrate social or psychological factors (eg., stress, family dietary habits)?
Response: To date, studies with integrated models are still scarce, and often do not take into account possible epigenetic factors such as diet, stress, environment, etc. We hope that this review can be a starting point for investigating these aspects in a holistic version.
Comment 3: Given the richness of the content, it may be helpful to include a few brief summary sentences or transitional paragraphs between sections to strengthen coherence and help readers follow the complex chains of biological processes more easily.
Response: We thank the reviewer for this observation and have added more "summary" key points at the end of the more complex paragraphs.
Reviewer 3 Report
Comments and Suggestions for Authors
Reviewer Report
Manuscript Title: Gut Microbiota and Autism: Unlocking Connections
Journal: Nutrients (2025)
Manuscript Type: Review Article
- General Assessment
This manuscript provides a comprehensive and up-to-date overview of the current understanding of the relationship between gut microbiota, diet, and neurodevelopment in Autism Spectrum Disorder (ASD). The topic is highly relevant, timely, and aligns with the scope of Nutrients. The authors successfully integrate mechanistic insights with clinical evidence, highlighting how microbiota-targeted nutritional interventions may influence neurodevelopmental outcomes. The writing is clear and well-structured, and the review demonstrates an impressive breadth of literature coverage.
However, while the manuscript is rich in content, it would benefit from some refinement in structure, focus, and critical analysis. The review sometimes reads as a detailed compilation of findings rather than a critical synthesis, and certain claims could be better supported or nuanced. There are also minor issues with formatting, citation consistency, and clarity in some sections.
- Major Comments
- Critical Synthesis vs. Descriptive Summary
The manuscript would benefit from deeper critical evaluation of conflicting findings and methodological limitations in the cited studies. For example, while numerous studies are mentioned linking dysbiosis to ASD, it is not always clear whether causality or correlation is established. A dedicated subsection discussing this issue and the challenges of establishing causality in microbiota–brain research would enhance the manuscript’s scientific rigor. - Structure and Flow
The review is extensive but occasionally repetitive, particularly between Sections 3 (Results) and 4 (Discussion). Consider merging or condensing overlapping content on the microbiota–gut–brain axis mechanisms. A visual summary (e.g., figure or table) showing the key mechanisms and pathways could help readers synthesize the material. - Clinical Relevance and Translational Perspective
While the authors highlight potential dietary and microbiota-targeted interventions, more critical discussion is needed on their clinical evidence and limitations. For instance, fecal microbiota transplantation (FMT), probiotics, and ketogenic diets are mentioned, but their long-term safety, standardization, and evidence quality should be addressed explicitly. - Methodology Section
The “Materials and Methods” section outlines the literature search but does not provide a PRISMA flow diagram or details on inclusion/exclusion criteria beyond general terms. Including these details would improve transparency and reproducibility. - Balance of Evidence
Some claims—particularly those linking maternal dysbiosis and environmental toxins directly to ASD—should be phrased more cautiously, emphasizing that evidence remains preliminary and largely associative. - Figure/Table Suggestions
Adding schematic illustrations (e.g., MGBA pathways, maternal microbiota effects, or dietary modulation strategies) and summary tables (key taxa changes, intervention outcomes) would substantially improve readability and educational value.
- Minor Comments
- Language and Style
The manuscript is generally well written, though certain sentences are overly long and could be simplified for clarity.
Example: Lines 236–243 (discussion on microglial activation) could be condensed and clarified. - References
There are several repeated or redundant references (e.g., citations 21–22 and 27–28). Please ensure each reference number is unique and consistently formatted according to Nutrients guidelines. - Terminology
Ensure consistent use of key terms such as “microbiota–gut–brain axis (MGBA)” and “gut–brain axis.” Some sections alternate between the two without definition. - Abstract
The abstract is informative but could be shortened by removing minor methodological details and emphasizing the novelty and conclusions. - Formatting
Some spacing and punctuation inconsistencies are present (e.g., double periods, missing spaces around citations). A careful proofread is recommended.
Author Response
Comments:
- Critical Synthesis vs. Descriptive Summary
The manuscript would benefit from deeper critical evaluation of conflicting findings and methodological limitations in the cited studies. For example, while numerous studies are mentioned linking dysbiosis to ASD, it is not always clear whether causality or correlation is established. A dedicated subsection discussing this issue and the challenges of establishing causality in microbiota–brain research would enhance the manuscript’s scientific rigor.
Response: We thank you for your constructive comments. We fully agree that a more thorough critical evaluation of the evidence, especially regarding discrepancies between studies and methodological limitations, could strengthen the overall quality of the manuscript. In response to your comment, we have added a dedicated subsection: “Current ASD and microbiota: limits and future prospects”.
Comments Structure and Flow
The review is extensive but occasionally repetitive, particularly between Sections 3 (Results) and 4 (Discussion). Consider merging or condensing overlapping content on the microbiota–gut–brain axis mechanisms. A visual summary (e.g., figure or table) showing the key mechanisms and pathways could help readers synthesize the material.
Response: Thank you for your comment. We fully accept your point regarding some redundancies between Sections 3 (Results) and 4 (Discussion). Following your comment, we have revised both sections, condensing overlapping content and improving the distinction between the presentation of results and their critical interpretation. Furthermore, as suggested, we have added a summary figure summarizing the main mechanisms and pathways involved in the microbiota–gut–brain interaction, to facilitate understanding and provide an integrated view of the data discussed.
Comments Clinical Relevance and Translational Perspective
While the authors highlight potential dietary and microbiota-targeted interventions, more critical discussion is needed on their clinical evidence and limitations. For instance, fecal microbiota transplantation (FMT), probiotics, and ketogenic diets are mentioned, but their long-term safety, standardization, and evidence quality should be addressed explicitly.
Response: We thank the reviewer and have added a dedicated section that critically examines the current limitations of microbiome interventions, both dietary and supplement-based, discussing the quality of available clinical evidence, issues of standardization and long-term safety, as well as outlining future prospects and the main remaining challenges.
Comments Methodology Section
The “Materials and Methods” section outlines the literature search but does not provide a PRISMA flow diagram or details on inclusion/exclusion criteria beyond general terms. Including these details would improve transparency and reproducibility.
Response: We thank the reviewer and we have added a PRISMA flow diagram in the section Materials and Methods
Comments Balance of Evidence
Some claims—particularly those linking maternal dysbiosis and environmental toxins directly to ASD—should be phrased more cautiously, emphasizing that evidence remains preliminary and largely associative.
Response: Thank you for your comment. We have reviewed the entire manuscript and highlighted the correlations, specifying that we are still far from being able to establish causal links.
Comments Figure/Table Suggestions
Adding schematic illustrations (e.g., MGBA pathways, maternal microbiota effects, or dietary modulation strategies) and summary tables (key taxa changes, intervention outcomes) would substantially improve readability and educational value.
Response: We fully accept your observation. We've added a figure and table that can help the reader get an overall picture.
Comments Minor Comments
- Language and Style
The manuscript is generally well written, though certain sentences are overly long and could be simplified for clarity.
Example: Lines 236–243 (discussion on microglial activation) could be condensed and clarified.
Response: Thank you for your comment. We have revised the entire manuscript to improve stylistic clarity, reducing sentence length and simplifying complex passages. In particular, the section on microglia (lines 236–243) has been rewritten to be more concise and clear, maintaining scientific precision but making it easier to read.
Comments References
There are several repeated or redundant references (e.g., citations 21–22 and 27–28). Please ensure each reference number is unique and consistently formatted according to Nutrients guidelines.
Response: We fully accept your comment. We have corrected all duplicate citations, including the cases you reported (21–22 and 27–28). All references are now unique, verified, and reformatted according to Nutrients journal guidelines.
Comments Terminology
Ensure consistent use of key terms such as “microbiota–gut–brain axis (MGBA)” and “gut–brain axis.” Some sections alternate between the two without definition.
Response: Thank you for highlighting the terminological inconsistency. We have now standardized the use of terms, consistently adopting the term "microbiota–gut–brain axis (MGBA)."
Comments Abstract
The abstract is informative but could be shortened by removing minor methodological details and emphasizing the novelty and conclusions.
Response: We accepted the suggestion and shortened the abstract by eliminating superfluous methodological details, emphasizing instead the most relevant elements, namely the novelty of the review and the main conclusions.
Comments Formatting
Some spacing and punctuation inconsistencies are present (e.g., double periods, missing spaces around citations). A careful proofread is recommended.
Response: Thank you for highlighting this. We have carefully reviewed the manuscript to correct all formatting inconsistencies, including double colons, missing spacing, and irregularities in punctuation and citations. The updated version now meets the required editorial standards.
Round 2
Reviewer 1 Report
Comments and Suggestions for Authors
The authors made quite big modifications and replied my comments. Now the quality is much better. Yet, the ending of the manuscript (Sections 9, 10, 11) seems endless. I suggest the authors reorganize them, reducing "1,2,3,4", using correct subtitle levels, and give a better logical flow.
Besides, the expression "microbiota–gut–brain axis" should be explained better. Because it is not a virtual axis, but an abstract concept (with a lot of fundemental studies). The authors may give more detailed discussion on this concept connecting to the current topic, autism.
Author Response
The authors made quite big modifications and replied my comments. Now the quality is much better. Yet, the ending of the manuscript (Sections 9, 10, 11) seems endless. I suggest the authors reorganize them, reducing "1,2,3,4", using correct subtitle levels, and give a better logical flow.
COMMENTS: We sincerely thank the reviewer for the positive evaluation and for acknowledging the substantial revisions made. We agree that Sections 9, 10, and 11 are currently too long and may appear fragmented. In the next revision, we will reorganize these sections to ensure a more coherent logical flow. Specifically, we have reduced the excessive enumeration (“1, 2, 3, 4”), and adjusted the hierarchy of subtitles to follow a consistent structure.
Besides, the expression "microbiota–gut–brain axis" should be explained better. Because it is not a virtual axis, but an abstract concept (with a lot of fundemental studies). The authors may give more detailed discussion on this concept connecting to the current topic, autism.
Comments 2: Regarding the expression “microbiota–gut–brain axis”, we appreciate the reviewer’s point. We have added a clearer explanation, and we have also elaborated on how this concept specifically relates to autism, discussing mechanistic pathways and current findings that connect microbiota alterations with neurodevelopmental outcomes. Once again, we thank the reviewer for the valuable comments, which will significantly enhance the manuscript.
Reviewer 3 Report
Comments and Suggestions for Authors
None.
Author Response
Thanks to the reviewer.
Best regards